# StructFormer: Joint Unsupervised Induction of Dependency and Constituency Structure from Masked Language Modeling

## Abstract

There are two major classes of natural language grammars — the dependency grammar that models one-to-one correspondences between words and the constituency grammar that models the assembly of one or several corresponded words. While previous unsupervised parsing methods mostly focus on only inducing one class of grammars, we introduce a novel model, StructFormer, that can induce dependency and constituency structure at the same time. To achieve this, we propose a new parsing framework that can jointly generates constituency tree and dependency graph. Then we integrate the induced dependency relations into transformer, in a differentiable manner, through a novel dependency-constrained self-attention mechanism. Experimental results show that our model can achieve strong results on unsupervised constituency parsing, unsupervised dependency parsing and masked language modeling at the same time.

## 1 Introduction

Human languages have a rich latent structure. This structure is multifaceted, with the two major classes of grammar being dependency and constituency structures. There have been an exciting breath of recent work that are targeted at learning this structure in a data-driven unsupervised fashion. The core principle behind recent methods that induce structure from data is simple - provide an inductive bias that is conducive for structure to emerge as a byproduct of some self-supervised training, e.g., language modeling. To this end, a wide range of models have been proposed that are able to successfully learn grammar structures (Shen et al., 2018a;c; Wang et al., 2019; Kim et al., 2019b;a). However, most of these works focus on learning constituency structures alone. To the best of our knowledge, there have been no prior model or work that is able to induce, in an unsupervised fashion, more than one grammar structure at once.

In this paper, we make two important technical contributions. First, we introduce the a new neural model that is able to induce dependency structures from raw data in an end-to-end unsupervised fashion. Most of existing approaches induce dependency structures from other syntactic information like gold POS tags (Klein & Manning, 2004; Cohen & Smith, 2009; Jiang et al., 2016). Previous works, that have trained from words alone, often requires additional information, like pre-trained word clustering (Spitkovsky et al., 2011), pre-trained word embedding (He et al., 2018), acoustic cues (Pate & Goldwater, 2013), or annotated data from related languages (Cohen et al., 2011). Second, we introduce the first neural model that is able to induce **both** dependency structure and constituency structure at the same time. Specifically, our approach aims to unify latent structure induction of different types of grammar within the same framework.

We introduce a new inductive bias that enables the Transformer models to induce a directed dependency graph in a fully unsupervised manner. To avoid the need of grammar labels during training, we use a distance-based parsing mechanism. The key idea is that it predicts a sequence of Syntactic Distances T (Shen et al., 2018b) and a sequence of Syntactic Heights $\Delta$ (Luo et al., 2019) to represent dependency graph and constituency trees at the same time. Examples of $\Delta$ and T are illustrated in Figure 1a. Based on the syntactic distances (T) and syntactic heights ($\Delta$), we provide a new dependency-constrained self-attention layer to replace the multi-head self-attention layer in standard transformer model. More concretely, each attention head can only attend on its parent (to avoid

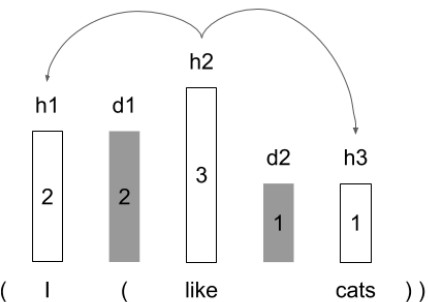

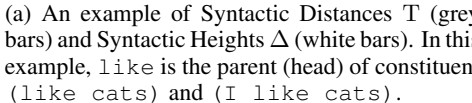

(a) An example of Syntactic Distances T (grey bars) and Syntactic Heights $\Delta$ (white bars). In this example, `like` is the parent (head) of constituent (`like cats`) and (`I like cats`).

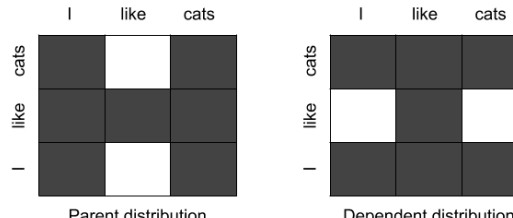

(b) Two types of dependency relations. The parent distribution allows each token to attend on its parent. The dependent distribution allows each token to attend on its dependents. For example the parent of `cats` is `like`. `Cats` and `I` are dependents of `like` Each attention head will receive a different weighted sum of these relations.

Figure 1: An example of our parsing mechanism and dependency-constrained self-attention mechanism. The parsing network first predicts the syntactic distance T and syntactic height $\Delta$ to represent the latent structure of the input sentence `I like cats`. Then the parent and dependent relations are computed in a differentiable manner from T and $\Delta$.

confusion with self-attention head, we use "parent" to note "head" in dependency graph) or its dependents in the predicted dependency structure, through a weighted sum of different relations shown in Figure 1b. In this way, we replace the complete graph in the standard transformer model with a differentiable directed dependency graph. During the process of training on a downstream task (e.g. masked language model), the model will gradually converge to a reasonable dependency graph via gradient descent. Thus, the parser can be trained in an unsupervised manner as a component of the model.

Incorporating the new parsing mechanism, the dependency-constrained self-attention, and the Transformer architecture, we introduce a new model named StructFormer. The proposed model can perform unsupervised dependency and constituency parsing at the same time, and can leverage the parsing results to achieve strong performance on masked language model tasks.

## 2 RELATED WORK

Previous works on unsupervised dependency parsing are primarily based on the dependency model with valence (DMV) (Klein & Manning, 2004) and its extension (Daumé III, 2009; Gillenwater et al., 2010). To effectively learn the DMV model for better parsing accuracy, a variety of inductive biases and handcrafted features, such as correlations between parameters of grammar rules involving different part-of-speech (POS) tags, have been proposed to incorporate prior information into learning. The most recent progress is the neural DMV model (Jiang et al., 2016), which uses a neural network model to predict the grammar rule probabilities based on the distributed representation of POS tags. However, most previous unsupervised dependency parsing algorithms require the gold POS tags as input, which are labeled by humans and can be potentially difficult (or prohibitively expensive) to obtain for large corpora. Spitkovsky et al. (2011) proposed to overcome this problem with unsupervised word clustering that can dynamically assign tags to each word considering its context.

Unsupervised constituency parsing has recently received more attention. PRPN (Shen et al., 2018a) and ON-LSTM (Shen et al., 2018c) induce tree structure by introducing an inductive bias to recurrent neural networks. PRPN proposes a parsing network to compute the syntactic distance of all word pairs, and a reading network utilizes the syntactic structure to attend to relevant memories. ON-LSTM allows hidden neurons to learn long-term or short-term information by a novel gating mechanism and activation function. In URNNG (Kim et al., 2019b), amortized variational inference was applied between a recurrent neural network grammar (RNNG) (Dyer et al., 2016) decoder and a tree structure inference network, which encourages the decoder to generate reasonable tree structures. DIORA (Drozdov et al., 2019) proposed using inside-outside dynamic programming to

compose latent representations from all possible binary trees. The representations of inside and outside passes from the same sentences are optimized to be close to each other. Compound PCFG (Kim et al., 2019a) achieves grammar induction by maximizing the marginal likelihood of the sentences which are generated by a probabilistic context-free grammar (PCFG) in a corpus. Tree Transformer (Wang et al., 2019) adds extra locality constraints to the Transformer encoder's self-attention to encourage the attention heads to follow a tree structure such that each token can only attend on nearby neighbors in lower layers and gradually extend the attention field to further tokens when climbing to higher layers.

Though large scale pre-trained models have dominated most natural language processing tasks, some recent work indicates that neural network models can see accurarcy gains by leveraging syntactic information rather then ignoring it (Marcheggiani & Titov, 2017; Strubell et al., 2018). Strubell et al. (2018) introduces syntactically-informed self-attention that force one attention head to attend on the syntactic governor of input token. Omote et al. (2019) and Deguchi et al. (2019) argue that dependency-informed self-attention can improve Transformer's performance on machine translation. Kuncoro et al. (2020) shows that syntactic biases help large scale pre-trained models, like BERT, to achieve better language understanding.

## 3 SYNTACTIC DISTANCE AND HEIGHT

In this section, we first reintroduce the concepts of syntactic distance and height, then discuss their relations in the context of StructFormer.

**Algorithm 1** Distance to binary constituency tree

1: **function** CONSTITUENT($\mathbf{w}$, $\mathbf{d}$)
2:     **if** $\mathbf{d} = []$ **then**
3:         $\mathbf{T} \Leftarrow \text{Leaf}(\mathbf{w})$
4:     **else**
5:         $i \Leftarrow \arg\max_i(\mathbf{d})$
6:         $\text{child}_l \Leftarrow \text{Constituent}(\mathbf{w}_{\leq i}, \mathbf{d}_{<i})$
7:         $\text{child}_r \Leftarrow \text{Constituent}(\mathbf{w}_{>i}, \mathbf{d}_{>i})$
8:         $\mathbf{T} \Leftarrow \text{Node}(\text{child}_l, \text{child}_r)$
9:     **return** $\mathbf{T}$

**Algorithm 2** Converting binary constituency tree to dependency graph

1: **function** DEPENDENT($\mathbf{T}$, $\Delta$)
2:     **if** $\mathbf{T} = w$ **then**
3:         $\mathbf{D} \Leftarrow []$, $\text{parent} \Leftarrow w$
4:     **else**
5:         $\text{child}_l, \text{child}_r \Leftarrow \mathbf{T}$
6:         $\mathbf{D}_l, \text{parent}_l \Leftarrow \text{Dependent}(\text{child}_l, \Delta)$
7:         $\mathbf{D}_r, \text{parent}_r \Leftarrow \text{Dependent}(\text{child}_r, \Delta)$
8:         $\mathbf{D} \Leftarrow \text{Union}(\mathbf{D}_l, \mathbf{D}_r)$
9:         **if** $\Delta(\text{parent}_l) > \Delta(\text{parent}_r)$ **then**
10:           $\mathbf{D}.\text{add}(\text{parent}_l \leftarrow \text{parent}_r)$
11:           $\text{parent} \Leftarrow \text{parent}_l$
12:         **else**
13:           $\mathbf{D}.\text{add}(\text{parent}_r \leftarrow \text{parent}_l)$
14:           $\text{parent} \Leftarrow \text{parent}_r$
15:     **return** $\mathbf{D}$, parent

### 3.1 SYNTACTIC DISTANCE

Syntactic distance is proposed in Shen et al. (2018b) to quantify the process of splitting sentences into smaller constituents.

**Definition 3.1.** Let $\mathbf{T}$ be a constituency tree for sentence $(w_0, ..., w_n)$. The height of the lowest common ancestor for consecutive words $x_i$ and $x_{i+1}$ is $\tilde{\tau}_i$. Syntactic distances $\mathbf{T} = (\tau_0, ..., \tau_{n-1})$ are defined as a sequence of $n-1$ real scalars that share the same rank as $(\tilde{\tau}_0, ..., \tilde{\tau}_{n-1})$.

In other words, each syntactic distance $d_i$ is associated with a split point $(i, i+1)$ and specify the relative order in which the sentence will be split into smaller components. Thus, any sequence of $n-1$ real values can unambiguously map to an unlabeled binary constituency tree with $n$ leaves through the Algorithm 1 (Shen et al., 2018b). As Shen et al. (2018c;a); Wang et al. (2019) pointed out, the syntactic distance reflects the information communication between constituents. More concretely, a large syntactic distance $\tau_i$ represents that less short-term or local information should be communicated between $(x_{\leq i})$ and $(x_{>i})$. While cooperating with correct inductive bias, we can leverage this feature to build unsupervised dependency parsing models.

## 3.2 Syntactic Height

Syntactic height is proposed in Luo et al. (2019), where the syntactic height is used to capture the distance to the root node in a dependency graph. A word with high syntactic height means it is close to the root node. In this paper, to match the definition of syntactic distance, we redefine syntactic height as:

**Definition 3.2.** Let $\mathbf{D}$ be a dependency graph for sentence $(w_0, ..., w_n)$. The height of a token $w_i$ in $\mathbf{D}$ is $\tilde{\delta}_i$. The syntactic heights of $\mathbf{D}$ can be any sequence of $n$ real scalars $\Delta = (\delta_0, ..., \delta_n)$ that share the same rank as $(\tilde{\delta}_0, ..., \tilde{\delta}_{n-1})$.

Although the syntactic height is defined based on the dependency structure, we cannot rebuild the original dependency structure just by syntactic heights, since there is no information about whether a token should be attached to the left side or the right side. However, given a unlabelled constituent tree, we can convert it into a dependency graph with the help of syntactic distance. The converting process is similar to the standard process of converting constituency treebank to dependency treebank (Gelbukh et al., 2005). Instead of using the constituent labels and POS tags to identify the parent of each constituent, we simply assign the token with largest syntactic height as the parent of each constituent. The converting algorithm is described in Algorithm 2. In Appendix A.1, we also proposed a joint algorithm, that takes T and $\Delta$ as inputs and output constituency tree and dependency graph at the same time.

## 3.3 The relation between Syntactic Distance and Height

As discussed previously, the syntactic distance controls information communication between the two side of the split point. The syntactic height quantifies the centrality of each token in the dependency graph. A token with large syntactic height tend to have more long-term dependency relations to connect different part of the sentence together. In StructFormer, we quantify the syntactic distance and height in the same scale. Given a split point $(i, i + 1)$ and it's syntactic distance $\delta_i$, only tokens $x_j$ with $\tau_j > \delta_i$ can have connections across the split point $(i, i + 1)$. Thus tokens with small syntactic height will be limited to mostly attend on near tokens.

## 4 StructFormer

In this section, we present the StructFormer model. Figure 2a shows the architecture of Struct-Former, which includes a parser network and a Transformer module. The parser network predicts T and $\Delta$, then passes them to a set of differentiable functions to generate dependency distributions. The Transformer module takes these distributions and the sentence as input to computes a contextual embedding for each position. The StructFormer can be trained in an end-to-end fashion on a Masked Language Model task. In this setting, the gradient back propagates through the relation distributions into the parser.

## 4.1 Parsing Network

As shown in Figure 2b, the parsing network takes word embeddings as input and feeds them into several convolution layers:

$$s_{l,i} = \tanh\left(\text{Conv}\left(s_{l-1,i-W}, s_{l-1,i-W+1}, ..., s_{l-1,i+W}\right)\right) \tag{1}$$

where $s_{l,i}$ is the output of $l$-th layer at $i$-th position, $s_{0,i}$ is the input embedding of token $w_i$, and $2W + 1$ is the convolution kernel size.

Given the output of the convolution stack $s_{N,i}$, we parameterize the syntactic distance T as:

$$\tau_i = \mathbf{W}_1^\tau \tanh\left(\mathbf{W}_2^\tau \begin{bmatrix} s_{N,i} \\ s_{N,i+1} \end{bmatrix} + b_2^\tau\right) + b_1^\tau \tag{2}$$

where $\delta_i$ is the contextualized distance for the $i$-th split point between token $w_i$ and $w_{i+1}$. The syntactic height $\Delta$ is parameterized in a similar way:

$$\delta_i = \mathbf{W}_1^\delta \tanh\left(\mathbf{W}_2^\delta s_{N,i} + b_2^\delta\right) + b_1^\delta \tag{3}$$

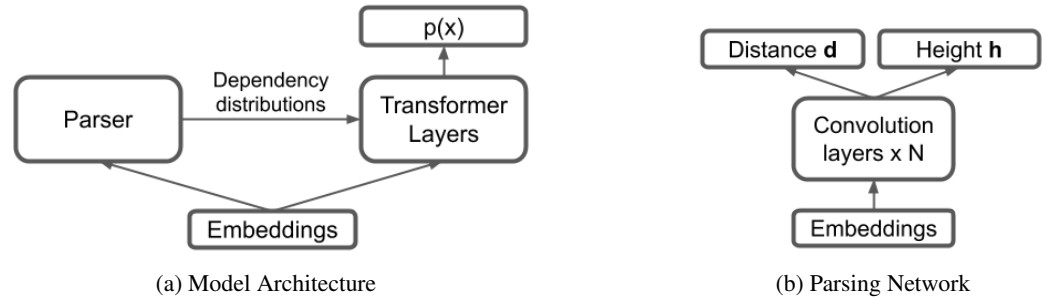

(a) Model Architecture                                          (b) Parsing Network

Figure 2: The Architecture of StructFormer. The parser takes shared word embeddings as input, outputs syntactic distances T, syntactic heights $\Delta$, and dependency distributions between tokens. The transformer layers take word embeddings and dependency distributions as input, output contextualized embeddings for input words.

## 4.2 ESTIMATE THE DEPENDENCY DISTRIBUTION

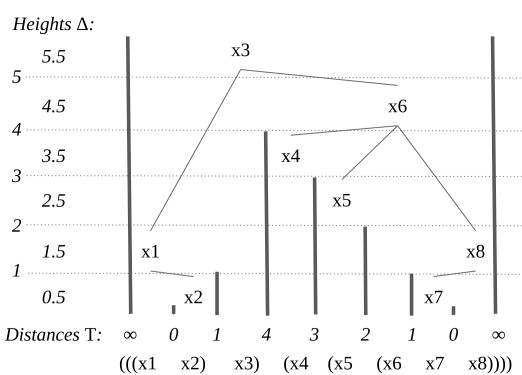

Figure 3: An example of T, $\Delta$ and respective dependency graph $\mathbf{D}$. $x_6$, thus $\mathbf{D}(x_4) = x_6$.

Given T and $\Delta$, we now explain how to estimate the probability $p(x_j|x_i)$ that the $j$-th token is the parent of the $i$-th token. The first step is identifying the smallest legit constituent $\mathbf{C}(x_i)$, that contains $x_i$ and $x_i$ is not $\mathbf{C}(x_i)$'s parent. The second step is identifying the parent of the constituent $x_j = \mathbf{Pr}(\mathbf{C}(x_i))$. Given the discussion in section 3.2, the parent of $\mathbf{C}(x_i)$ must be the parent of $x_i$. Thus, the two-stages of identifying the parent of $x_i$ can be formulated as:

$$\mathbf{D}(x_i) = \mathbf{Pr}(\mathbf{C}(x_i)) \qquad (4)$$

In StructFormer, $\mathbf{C}(x_i)$ is represented as constituent $[l, r]$, where $l$ is the starting index ($l \leq i$) of $\mathbf{C}(x_i)$ and $r$ is the ending index ($r \geq i$) of $\mathbf{C}(x_i)$. For example, in Figure 3, $\mathbf{C}(x_4) = [4, 8]$ and the parent of constituent $\mathbf{Pr}([4, 8]) = x_6$, thus $\mathbf{D}(x_4) = x_6$.

In a dependency graph, $x_i$ is only connected to its parent and dependents. It means that $x_i$ don't have direct connection to the outside of $\mathbf{C}(x_i)$. In other words, $\mathbf{C}(x_i) = [l, r]$ is the smallest constituent that satisfies:

$$\delta_i < \tau_{l-1}, \quad \delta_i < \tau_r \qquad (5)$$

where $\tau_{l-1}$ is the first $\tau_{<i}$ that is larger then $\delta_i$ while looking backward, and $\tau_r$ is the first $\tau_{\geq i}$ that is larger then $\delta_i$ while looking forward. In the previous, $\delta_4 = 3.5$, $\tau_3 = 4 > \delta_4$ and $\tau_8 = \infty > \delta_4$, thus $\mathbf{C}(x_4) = [4, 8]$. To make this process differentiable, we define $\tau_k$ as a real value and $\delta_i$ as a probability distribution $p(\tilde{\delta}_i)$. For the simplicity and efficiency of computation, we directly parameterize the cumulative distribution function $p(\tilde{\delta}_i > \tau_k)$ with sigmoid function:

$$p(\tilde{\delta}_i > \tau_k) = \sigma((\delta_i - \tau_k)/\mu_1) \qquad (6)$$

where $\sigma$ is the sigmoid function, $\delta_i$ is the mean of distribution $p(\tilde{\delta}_i)$ and $\mu_1$ is a learnable temperature term. Thus the probability that the $l$-th ($l < i$) token is inside $\mathbf{C}(x_i)$ is equal to the probability that $\tilde{\delta}_i$ is larger then the maximum distance $\tau$ between $l$ and $i$:

$$p(l \in \mathbf{C}(x_i)) = p(\tilde{\delta}_i > \max(\tau_{i-1}, ..., \tau_l)) = \sigma((\delta_i - \max(\tau_l, ..., \tau_{i-1}))/\mu) \qquad (7)$$

Then we can compute the probability distribution for $l$:

$$p_{left}(l|i) = \sum_{k \in [1,l]} p_{left}(k|i) - \sum_{k \in [1,l-1]} p_{left}(k|i) = p(l \in \mathbf{C}(x_i)) - p(l - 1 \in \mathbf{C}(x_i))$$

$$= \sigma((\delta_i - \max(\tau_l, ..., \tau_{i-1}))/\mu) - \sigma((\delta_i - \max(\tau_{l-1}, ..., \tau_{i-1}))/\mu) \qquad (8)$$

Similarly, we can compute the probability distribution for $r$:

$$p_{right}(r|i) = \sigma((\delta_i - \max(\tau_i, ..., \tau_{r-1}))/\mu) - \sigma((\delta_i - \max(\tau_i, ..., \tau_r))/\mu) \qquad (9)$$

The probability distribution for $[l, r] = \mathbf{C}(x_i)$ can be computed as:

$$p_{\mathbf{C}}([l, r]|i) = \begin{cases} p_{left}(l|i)p_{right}(r|i), & l \leq i \leq r \\ 0, & \text{otherwise} \end{cases} \qquad (10)$$

The second step is to identify the parent of $[l, r]$. For any constituent $[l, r]$, we choose the $j = \text{argmax}_{k \in [l,r]}(\delta_k)$ as the parent of $[l, r]$. In the previous example, given constituent $[4, 8]$, the maximum syntactic height is $\delta_6 = 4.5$, thus $\mathbf{Pr}([4, 8]) = x_6$. We use softmax function to parameterize the probability $p_{\mathbf{Pr}}(j|[l, r])$:

$$p_{\mathbf{Pr}}(j|[l, r]) = \begin{cases} \frac{\exp(h_j/\mu_2)}{\sum_{l \leq k \leq r} \exp(h_k/\mu_2)}, & l \leq t \leq r \\ 0, & \text{otherwise} \end{cases} \qquad (11)$$

Given probability $p(j|[l, r])$ and $p([l, r]|i)$, we can compute the probability that $x_j$ is the parent of $x_i$:

$$p_{\mathbf{D}}(j|i) = \begin{cases} \sum_{[l,r]} p_{\mathbf{Pr}}(j|[l, r])p_{\mathbf{C}}([l, r]|i), & i \neq j \\ 0, & i = j \end{cases} \qquad (12)$$

### 4.3 DEPENDENCY-CONSTRAINED MULTI-HEAD SELF-ATTENTION

The multi-head self-attention in transformer can be seen as a information propagation mechanism on the complete graph $\mathbf{G} = (X, E)$, where the set of vertices $X$ contains all $n$ tokens in the sentence, and the set of edges $E$ contains all possible word pairs $(x_i, x_j)$. StructFormer replace the complete graph $\mathbf{G}$ with a soft dependency graph $\mathbf{D} = (X, A)$, where $A$ is the set of $n$ probability distribution $\{p_{\mathbf{D}}(j|i)\}$ that represent the probability of existing and directed edge between the dependent $i$ and the parent $j$. The reason that we called it a directed edge is that each specific head is only allow to propagate information either from parent to dependent or from from dependent to parent. To do so, structformer associate each attention head with a probability distribution over parent or dependent relation.

$$p_{\text{parent}} = \frac{\exp(w_{\text{parent}})}{\exp(w_{\text{parent}}) + \exp(w_{\text{dep}})}, \quad p_{\text{dep}} = \frac{\exp(w_{\text{dep}})}{\exp(w_{\text{parent}}) + \exp(w_{\text{dep}})} \qquad (13)$$

where $w_{\text{parent}}$ and $w_{\text{dep}}$ are learnable parameters that associated with each attention head, $p_{\text{parent}}$ is the probability that this head will propagate information from parent to dependent, vice versa. The model will learn to assign this association from the downstream task via gradient descent. Then we can compute the probability that information can be propagated from node $j$ to node $i$ via this head:

$$p_{i,j} = p_{\text{parent}}p_{\mathbf{D}}(j|i) + p_{\text{dep}}p_{\mathbf{D}}(i|j) \qquad (14)$$

However, Htut et al. (2019) pointed out that different heads tend to associate with different type of universal dependency relations (including `nsubj`, `obj`, `advmod`, etc), but there is no generalist head can that work with all different relations. To accommodate this observation, we compute a individual probability for each head and pair of tokens $(x_i, x_j)$:

$$q_{i,j} = \text{sigmoid}\left(\frac{QK^T}{\sqrt{d_k}}\right) \qquad (15)$$

where $Q$ and $K$ are query and key matrix in a standard transformer model and $d_k$ is the dimension of attention head. The equation is inspired by the scaled dot-product attention in transformer. We replace the original softmax function with sigmoid function, so $q_{i,j}$ became an independent probability that indicate whether the specific could work with the work $(x_i, x_j)$. In the end, we propose to replace transformer's scaled dot-product attention with our dependency-constrained self-attention:

$$\text{Attention}(Q_i, K_j, V_j, \mathbf{D}) = p_{i,j}q_{i,j}V_j \qquad (16)$$

## 5 EXPERIMENTS

We evaluate the proposed model on three tasks: Masked Language Modeling, Unsupervised Consituency Parsing and Unsupervised Dependency Parsing.

Our implementation of StructFormer is close to the original Transformer encoder (Vaswani et al., 2017). Except that we put the layer normalization in front of each layer, similar to the T5 model (Raffel et al., 2019). We found that this modification allows the model to converges faster. For all experiments, we set the number of layers $L = 8$, the embedding size and hidden size to be $d_{model} = 512$, the number of self-attention heads $h = 8$, the feed-forward size $d_{ff} = 2048$, dropout rate as $0.1$, and the number of convolution layers in the parsing network as $L_p = 3$.

### 5.1 MASKED LANGUAGE MODEL

Masked Language Modeling (MLM) has been widely used as pretraining object for larger scale pre-training models. In BERT (Devlin et al., 2018) and RoBERTa (Liu et al., 2019), authors found that MLM perplexities on held-out evaluation set have a positive correlation with the end-task performance. We trained and evaluated our model on 2 different datasets: the Penn TreeBank (PTB) and BLLIP. In our MLM experiments, each individual token has a independent chance to be replaced by a mask token <mask>, except that we never replace $< unk >$ token. The training and evaluation object for Masked Language Model is to predict the replaced tokens. The performance of MLM is evaluated measuring perplexity on masked words.

**PTB** is a standard dataset for language modeling (Mikolov et al., 2012) and unsupervised constituency parsing (Shen et al., 2018c; Kim et al., 2019a). Following the setting proposed in Shen et al. (2018c), we use Mikolov et al. (2012)'s prepossessing process, which removes all punctuations, and replaces low frequency tokens with <unk>. The preprocessing results in a vocabulary size of 10001 (including <unk>, <pad> and <mask>). For PTB, we use a 30% mask rate.

**BLLIP** is a large Penn Treebank-style parsed corpus of approximately 24 million sentences. We train and evaluate StructFormer on three splits of BLLIP: BLLIP-XS (40k sentences, 1M tokens), BLLIP-SM (200K sentences, 5M tokens), and BLLIP-MD (600K sentences, 14M tokens). They are obtained by randomly sampling sections from BLLIP 1987-89 Corpus Release 1. All models are tested on a shared held-out test set (20k sentences, 500k tokens). Following the settings provided in (Hu et al., 2020), we use subword-level vocabulary extracted from the GPT-2 pre-trained model rather than the BLLIP training corpora. For BLLIP, we use a 15% mask rate.

### 5.2 UNSUPERVISED CONSTITUENCY PARSING

The unsupervised constituency parsing task compares the latent tree structure induced by the model with those annotated by human experts. We use the Alogrithm 1 to predict the constituency trees from T predicted by StructFormer. Following the experiment settings proposed in Shen et al. (2018c), we take the model trained on PTB dataset, and evaluate it on WSJ test set. The WSJ test set is the section 23 of WSJ corpus, it contains 2416 human expert labeled sentences. Punctuation is ignored during the evaluation.

### 5.3 UNSUPERVISED DEPENDENCY PARSING

The unsupervised dependency parsing evaluation compares the induced dependency relations with those in the reference dependency graph. The most common metric is Unlabeled Attachment Score (UAS), which measures the percentage that a token is correctly attached to its parent in the reference tree. Another widely used metric for unsupervised dependency parsing is Undirected Unlabeled Attachment Score (UUAS) measures the percentage that the reference undirected and unlabeled connections are recovered by the induced tree. Similar to the unsupervised constituency parsing, we take the model trained on PTB dataset, and evaluate it on WSJ test set (section 23). For the WSJ test set, reference dependency graphs are converted from its human annotated constituency trees. However, there are two different sets of rules for the conversion: the Stanford dependencies and the CoNLL dependencies. While Stanford dependencies are used as reference dependencies in previous unsupervised parsing papers, we noticed that our model sometimes output dependency structures that are closer to the CoNLL dependencies. Therefore, we report UAS and UUAS for both Stanford

| Model | PTB | BLLIP-XS | BLLIP-SM | BLLIP-MD |
|---|---|---|---|---|
| Transformer | 64.05 | 93.90 | 19.92 | 14.31 |
| StructFormer | 60.94 | 57.28 | 18.70 | 13.70 |

Table 1: Masked Language Model perplexities on BLLIP datasets of different models.

| Methods | UF1 |
|---|---|
| RANDOM | 21.6 |
| LBRANCH | 9.0 |
| RBRANCH | 39.8 |
| *Pre-trained LMs* | |
| BERT-large* | 34.2 |
| GPT2* | 37.1 |
| XLNet-base* | 40.1 |
| *Unsupervised Parsing Models* | |
| PRPN (Shen et al., 2018a) | 37.4 (0.3) |
| ON-LSTM (Shen et al., 2018c) | 47.7 (1.5) |
| Tree-T (Wang et al., 2019) | 49.5 |
| URNNG (Kim et al., 2019b) | 52.4 |
| C-PCFG (Kim et al., 2019a) | 55.2 |
| StructFormer | 54.0 (0.3) |

(a) Constituency Parsing Results. * results are from Kim et al. (2020).

| Methods | UAS |
|---|---|
| *w/o gold POS tags* | |
| DMV (Klein & Manning, 2004) | 35.8 |
| E-DMV (Headden III et al., 2009) | 38.2 |
| UR-A E-DMV (Tu & Honavar, 2012) | 46.1 |
| CS* (Spitkovsky et al., 2013) | 64.4* |
| Neural E-DMV (Jiang et al., 2016) | 42.7 |
| Gaussian DMV (He et al., 2018) | 43.1 (1.2) |
| INP (He et al., 2018) | 47.9 (1.2) |
| StructFormer | 46.2 (0.4) |
| *w/ gold POS tags (for reference only)* | |
| DMV (Klein & Manning, 2004) | 39.7 |
| UR-A E-DMV (Tu & Honavar, 2012) | 57.0 |
| MaxEnc (Le & Zuidema, 2015) | 65.8 |
| Neural E-DMV (Jiang et al., 2016) | 57.6 |
| CRFAE (Cai et al., 2017) | 55.7 |
| L-NDMV† (Han et al., 2017) | 63.2 |

(b) Dependency Parsing Results on WSJ testset. Starred entries (*) benefit from additional punctuation-based constraints. Daggered entries (†) benefit from larger additional training data. Baseline results are from He et al. (2018).

Table 2: The unsupervised parsing performance of different models.

and CoNLL dependencies. Following the setting of previous papers (Jiang et al., 2016), we ignored the punctuation during evaluation. To obtain the dependency relation from our model, we compute the argmax for dependency distribution:

$$k = \text{argmax}_{j \neq i} p_{\mathbf{D}}(j|i) \tag{17}$$

and assign the $k$-th token as the parent of $i$-th token.

## 5.4 EXPERIMENTAL RESULTS

The masked language model results are shown in Table 1. StructFormer consistently outperform our Transformer baseline. This result aligns with previous observations that linguistic informed self-attention can help Transformers achieve stronger performance. We also observe that StructFormer converges much faster than the standard Transformer model.

Table 2a shows that our model achieves strong results on unsupervised constituency parsing. While the C-PCFG (Kim et al., 2019a) achieve a stronger parsing performance with its strong linguistic constraints (e.g. a finite set of production rules), StructFormer may have border domain of application. For example, it can replace standard transformer encoder in most of popular large-scale pretrained language models (e.g. BERT and ReBERTa) and transformer based machine translation models. It's also interesting to notice that, different from Tree-T (Wang et al., 2019), we didn't directly use constituents to restrict the self-attention receptive field. But we eventually achieve a stronger constituency parsing performance with same experiment setting. This result may suggest that the dependency relations is a more suitable for grammar induction in transformer-based models. Table 3 shows that our model achieve strong accuracy while predicting Noun Phrase (NP), Preposition Phrase (PP), Adjective Phrase (ADJP), and Adverb Phrase (ADVP).

Table 2b shows that our model achieves competitive dependency parsing performance while comparing to other models that do not require gold POS tags. While most of baseline models still relies

|      | PRPN  | ON    | C-PCFG | Tree-T | StructFormer |
|------|-------|-------|--------|--------|--------------|
| SBAR | 50.0% | 52.5% | **56.1%** | 36.4% | 48.7% |
| NP   | 59.2% | 64.5% | **74.7%** | 67.6% | 72.1% |
| VP   | **46.7%** | 41.0% | 41.7% | 38.5% | 43.0% |
| PP   | 57.2% | 54.4% | 68.8% | 52.3% | **74.1%** |
| ADJP | 44.3% | 38.1% | 40.4% | 24.7% | **51.9%** |
| ADVP | 32.8% | 31.6% | 52.5% | 55.1% | **69.5%** |

Table 3: Fraction of ground truth constituents that were predicted as a constituent by the models broken down by label (i.e. label recall)

| Relations | MLM PPL | Constituency UF1 | Stanford | | Conll | |
|-----------|---------|------------------|----------|------|-------|------|
|           |         |                  | UAS      | UUAS | UAS   | UUAS |
| parent+dep | 60.9 (1.0) | 54.0 (0.3) | 46.2 (0.4) | 61.6 (0.4) | 36.2 (0.1) | 56.3 (0.2) |
| parent     | 63.0 (1.2) | 40.2 (3.5) | 32.4 (5.6) | 49.1 (5.7) | 30.0 (3.7) | 50.0 (5.3) |
| dep        | 63.2 (0.6) | 51.8 (2.4) | 15.2 (18.2) | 41.6 (16.8) | 20.2 (12.2) | 44.7 (13.9) |

Table 4: The performance of StructFormer with different combinations of attention masks.

on some kind of latent POS tags or pretrained word embeddings, StructFormer can be seen as a easy-to-use alternative that works in an end-to-end fashion. Table 5 shows that our model recovers 61.6% of undirected dependency relations. Given the strong performances on both dependency parsing and masked language modeling, we believe that the dependency graph schema could be an viable substitute for the complete graph schema used in standard transformer.

Since our model uses a mixture of relation probability distribution for each self-attention head, we also studied how different combinations of relations effect the performance of our model. Table 5 shows that the model can achieve the best performance, while using both parent and dependent relations. The model suffers more on dependency parsing, if the parent relation is removed. And if the dependent relation is removed, the model will suffers more on the constituency parsing.

## 6 CONCLUSION

In this paper, we introduce a novel dependency and constituency joint parsing framework. Based on the framework, we propose StructFormer, a new unsupervised parsing algorithm that does unsupervised dependency and constituency parsing at the same time. We also introduced a novel dependency-constrained self-attention mechanism that allows each attention head to focus on a specific mixture of dependency relations. This brings Transformers closer to modeling a directed dependency graph. The experiments show premising results that StructFormer can induce meaningful dependency and constituency structures and achieve better performance on masked language model task. This research provides a new path to build more linguistic bias into pre-trained language model.

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

# A  APPENDIX

## A.1  JOINT DEPENDENCY AND CONSTITUENCY PARSING

---

**Algorithm 3** The joint dependency and constituency parsing algorithm. Inputs are a sequence of words $\mathbf{w}$, syntactic distances $\mathbf{d}$, syntactic heights $\mathbf{h}$. Outputs are a binary constituency tree $\mathbf{T}$, a dependency graph $\mathbf{D}$ that is represented as a set of dependency relations, the parent of dependency graph $\mathbf{D}$, and the syntactic height of parent.

---

1: **function** BUILDTREE($\mathbf{w}, \mathbf{d}, \mathbf{h}$)
2:   **if** $\mathbf{d} = []$ and $\mathbf{w} = [w]$ and $\mathbf{h} = [h]$ **then**
3:     $\mathbf{T} \Leftarrow \text{Leaf}(w)$, $\mathbf{D} \Leftarrow []$, parent $\Leftarrow w$, height $\Leftarrow h$
4:   **else**
5:     $i \Leftarrow \arg\max(\mathbf{d})$
6:     $\mathbf{T}_l, \mathbf{D}_l, \text{parent}_l, \text{height}_l \Leftarrow \text{BuildTree}(\mathbf{d}_{<i}, \mathbf{w}_{\leq i}, \mathbf{h}_{\leq i})$
7:     $\mathbf{T}_r, \mathbf{D}_r, \text{parent}_r, \text{height}_r \Leftarrow \text{BuildTree}(\mathbf{d}_{>i}, \mathbf{w}_{>i}, \mathbf{h}_{>i})$
8:     $\mathbf{T} \Leftarrow \text{Node}(\text{child}_l \Leftarrow \mathbf{T}_l, \text{child}_r \Leftarrow \mathbf{T}_r)$
9:     $\mathbf{D} \Leftarrow \text{Union}(\mathbf{D}_l, \mathbf{D}_r)$
10:    **if** $\text{height}_l > \text{height}_r$ **then**
11:      $\mathbf{D}.\text{add}(\text{parent}_l \leftarrow \text{parent}_r)$
12:      parent $\Leftarrow \text{parent}_l$, height $\Leftarrow \text{height}_l$
13:    **else**
14:      $\mathbf{D}.\text{add}(\text{parent}_r \leftarrow \text{parent}_l)$
15:      parent $\Leftarrow \text{parent}_r$, height $\Leftarrow \text{height}_r$
16:   **return** $\mathbf{T}$, $\mathbf{D}$, parent, height

---

## B  DEPENDENCY RELATION WEIGHTS FOR SELF-ATTENTION HEADS

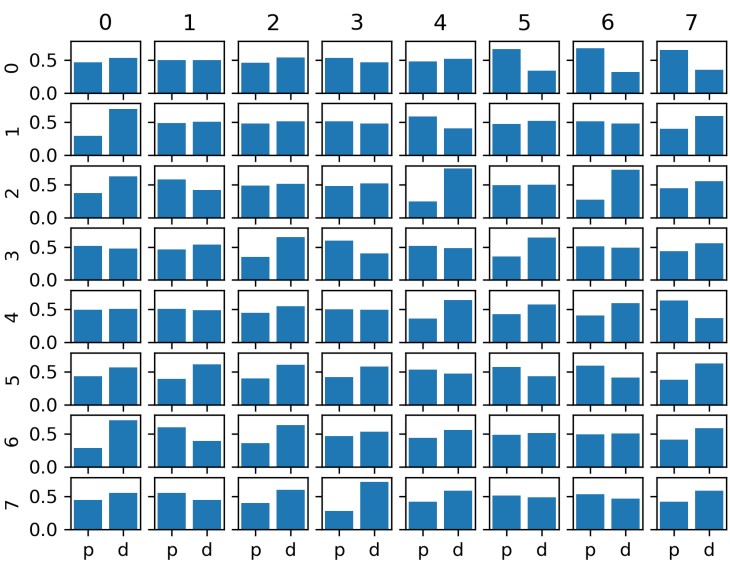

(a) Dependency relation weights learnt on PTB

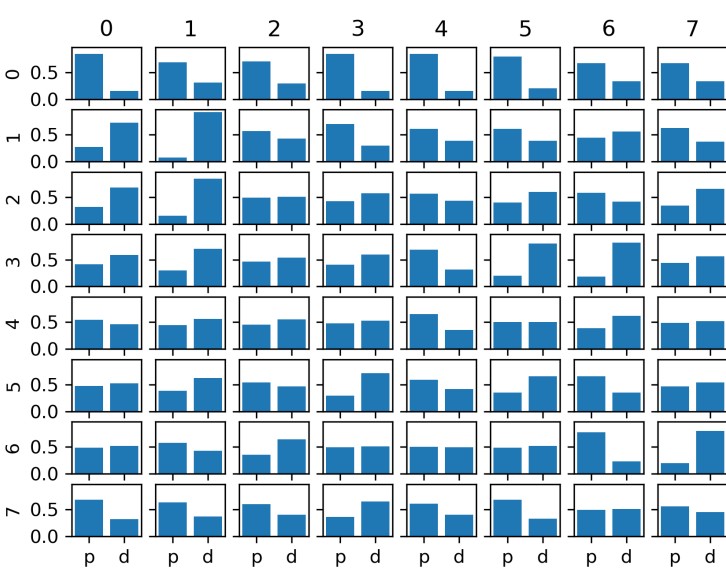

(b) Dependency relation weights learnt on BLLIP-SM

Figure 4: Dependency relation weights learnt on different datasets. Row $i$ constains relation weights for all attention heads in the $i$-th transformer layer. p represents the parent relation. d represents the dependent relation. We observe a clearer preference for each attention head in the model trained on BLLIP-SM. This probably due to BLLIP-SM has signficantly more training data. It's also interesting to notice that the first layer tend to focus on parent relations.

## C  DEPENDENCY DISTRIBUTION EXAMPLES

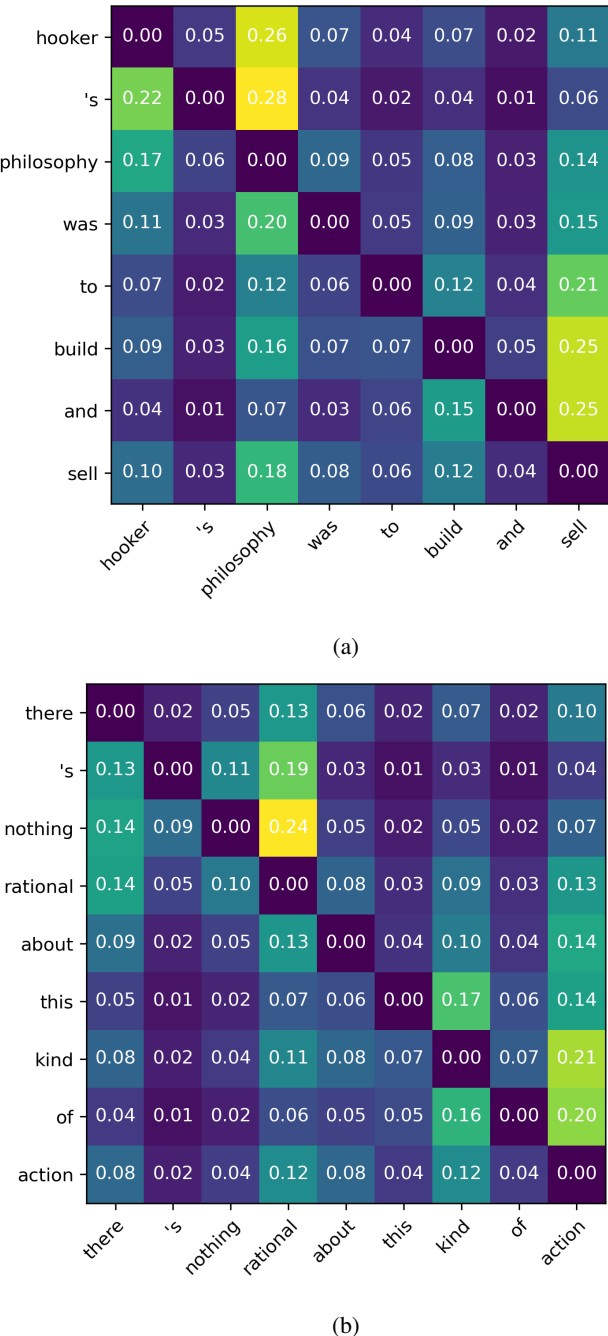

(a)

(b)

Figure 5: Dependency distribution examples from WSJ test set. Each row is the parent distribution for the respective word. The sum of each distribution may not equal to 1.

# D    THE PERFORMANCE OF STRUCTFORMER WITH DIFFERENT MASK RATES

| Mask rate | MLM PPL | Constituency UF1 | Stanford UAS | Stanford UUAS | Conll UAS | Conll UUAS |
|---|---|---|---|---|---|---|
| 0.1 | 45.3 (1.2) | 51.45 (2.7) | 31.4 (11.9) | 51.2 (8.1) | 32.3 (5.2) | 52.4 (4.5) |
| 0.2 | 50.4 (1.3) | 54.0 (0.6) | 37.4 (12.6) | 55.6 (8.8) | 33.0 (5.7) | 53.5 (4.7) |
| 0.3 | 60.9 (1.0) | 54.0 (0.3) | 46.2 (0.4) | 61.6 (0.4) | 36.2 (0.1) | 56.3 (0.2) |
| 0.4 | 76.9 (1.2) | 53.5 (1.5) | 34.0 (10.3) | 52.0 (7.4) | 29.5 (5.4) | 50.6 (4.1) |
| 0.5 | 100.3 (1.4) | 53.2 (0.9) | 36.3 (9.8) | 53.6 (6.8) | 30.6 (4.2) | 51.3 (3.2) |

Table 5: The performance of StructFormer on PTB dataset with different mask rates. Dependency parsing is especially affected by the masks. Mask rate 0.3 provides the best and the most stable performance.

