# OpenReview forum: "StructFormer: Joint Unsupervised Induction of Dependency and Constituency Structure from Masked Language Modeling"
_ICLR.cc/2021/Conference — Reject_

### Official Review · AnonReviewer3 · 2020-10-27
**interesting ideas, but needs a lot of work**

**Rating:** 4
**Confidence:** 4

**Review:**

Updates after discussion/revision period:

It appears that the paper has improved. However, the changes appear to be so substantial that the paper is now essentially a different paper which would require a new review process.

---------

This paper describes a neural architecture that resembles the transformer but includes explicit representations of constituency and dependency structure. The model is trained for masked language modeling (MLM) and evaluated via MLM on held-out data and its ability to induce constituency and dependency trees. The results are better than trivial baselines for unsupervised constituency and dependency parsing but not as strong as related recent work.

This paper has an interesting idea at its core: stemming from the success of models like ordered neurons, can we define a neural architecture that uses both constituency and dependency structure? The related notions of height and distance, combined with the syntactically-inspired attention masks, also seem interesting.

Unfortunately, however, the current submission needs a lot of work before it will be ready for publication. There are three types of concerns I have about the paper in its current form:

1. The description of the method is dense and difficult to understand, and has at least a few notational issues (details below).

2. There are very few experimental details provided. Yes, space is limited in the main paper, but appendices are permitted as well and there were no appendices included in the submission. Basic questions about the experiments are not included, making it difficult to assess the paper.

3. The experimental results are difficult to evaluate, as there are different datasets and assumptions made by different work, so there does not appear to be a baseline that the new model can be compared to fairly. it's unclear what the take-home message is. It would be beneficial to have some simpler baselines run by the authors on the same data they are using for training, in order to have a comparison that is fair.

I'll discuss these three points below.

 1. Clarity of model description and notational issues.
First, I had some confusion about Alg. 1.
- The BuildTree function call in line 8 does not match the function signature in line 1. Or is it supposed to be calling some other function that's not part of this algorithm?
- What is k on line 12? k only seems to be defined in line 13.
- I don't understand the notation in this part of line 14: "head_i \in {head_i}\head_k". I think it's confusing to use i in both head_i and {head_i}, because i is (I think) being used as an index which is iterated over. It would be more clear to use a different index (e.g., j) for one of these two instances of i.

I also had some confusion about the StructFormer description.
- Eq. (5) contains p_j^{C(i)} but then right below it when the equation is being described, there is a p_{i,j}^{ct} -- are these two related in some way? It's not clear to me how the notation should line up here.
- Why does Eq. (9) have p(i \leftarrow k) repeated twice? I think one of them should have a j.

In addition to my confusion about the uncertainties above, I found it difficult to follow the description of the StructFormer due to its density. Sec 4 would benefit from a more leisurely exposition, along with figures and potentially an example.

2. The experiments are difficult for me to assess because many details are missing. For example:
(a) How is masking done during training with masked language modeling? That is, what is the masking rate? Are whole words masked or subword units? Are any words that were selected to be masked left unchanged or swapped to be other words as in BERT?
(b) What optimizer was used and how were learning rates selected? Was any early stopping done? If so, what criterion was used?
(c) What is the value of the convolution kernel parameter W?
(d) How was the gap value of 0.1 selected? How sensitive is the performance to this value?
(e) It's not clear to me whether the masked language modeling comparison between the transformer and structformer is fair. Is the number of parameters comparable between those two models? Were they trained in similar ways?
(f) UUAS is never actually defined (though I was able to figure it out based on context).

3. The empirical comparison leaves something to be desired. The StructFormer reaches higher parsing accuracies than trivial baselines, but not as high as related recent work. However, the paper notes that it does not consider that related work as perfectly comparable. Some results that I think would be more comparable are not included in the results tables. Page 1 has the following text: "Previous works, that have trained from words alone, often requires additional information, like pre-trained word clustering (Spitkovsky et al., 2011), pre-trained word embedding (He et al., 2018),...".  Pretrained clusters or embeddings are unsupervised and therefore would be reasonable to compare to. The results tables omit results from these papers, instead reporting results from papers that use gold POS tags and other such information. But I would argue that a system that uses unsupervised word clusters or unsupervised word embeddings is fair to compare to. Why aren't the results from Spitkovsky et al and He et al included in the results tables?  In addition to comparing to prior work, it would help to characterize the performance of the StructFormer to include baselines that are trained on the same dataset as that used by the authors, so that we can disentangle the impact of some of the choices made here.


Below are typos and minor things:

- p. 5: "higher then all" --> "higher than all"
- p. 5: "a unsupervised" --> "an unsupervised"
- p. 6: "to converges faster" --> "to converge faster"
- p. 6: "Alogrithm" --> "Algorithm"
- p. 7: The sentence starting with "Since POS tags" is not a complete sentence.
- p. 7: "infering" --> "inferring"
- p. 7: "effect" --> "affect"
- p. 7: The sentence starting with "Because it is" is not a complete sentence.
- p. 7: "only have" --> "only having"
- p. 8: The caption of Fig 4 mentions s, but I think c was meant instead, as c is present in the figure while s is not.
- p. 8: "premising" --> "promising"

---

### Official Review · AnonReviewer4 · 2020-10-28
**Interesting architecture, but numerous concerns regarding evaluation and exposition**

**Rating:** 4
**Confidence:** 3

**Review:**

Summary: this paper introduces a new deep-learning architecture for unsupervised parsing.  From a CNN on the input sentence, the model predicts a syntactic "height" for every word in the sentence and a syntactic "distance" for every inter-word position; these heights and distances determine the joint constituency & dependency parse of the sentence.  The parse in turn determines a soft attention mask for a Transformer encoder module.  The model is trained end-to-end on masked language modeling (MLM).  The model shows improvements in MLM perplexity over a standard Transformer baseline, and the authors present it as offering competitive unsupervised parsing scores relative to other models.

Evaluation: the architecture is interesting and different enough to my knowledge from previous architectures that have been proposed that, together with competitive results on unsupervised parsing, would make for a compelling submission.  However, there are a number of unclarities in the paper that make it difficult to determine whether the results on unsupervised parsing are actually comparable to prior work.  It is essential that these unclarities be satisfactorily addressed in order to determine whether the paper is actually a candidate for acceptance.  I list the issues below; I am open to revising my score more positively if the authors are able to provide a compelling response.

* The start of Section 5 states that the model is trained on BLLIP datasets and unsupervised parsing evaluation is on the WSJ testset.  Sections 5.2 and 5.3 don't say anything about using a different training set (e.g., the standard WSJ training set sections 2–21 as in Kim et al. 2019a), so I assume that the present paper is using BLLIP as a training set here (though the paper doesn't say which BLLIP dataset).  If that's the case, then unless the authors have re-trained all the competitor models listed in Table 1 on the BLLIP dataset, the StructFormer results are not comparable to any prior work because the training dataset isn't the same.  If I'm correct about this, it constitutes clear grounds for rejection of the paper.

* The dependency results in Figure 1b are also confusing.  RANDOM and RBRANCH/LHEAD results are implausibly low (RBRANCH/LHEAD should be better than LBRANCH/RHEAD for English).  I don't see where the DMV or Shared LN results come from either.  Note also that Klein & Manning 2004 include results that don't use gold POS tags but rather word classes using a simple distributional clustering technique, and DMV takes only a fairly small performance hit from this, so it is not true that none of the * models can be directly compared with StructFormer.

* Table 3 doesn't seem consistent with Table 1b: none of the dependency results in Table 3 match the 41.0 result for StructFormer listed in 1b.  Likewise, Table 4 MLM perplexities aren't consistent with Table 2.

* Section 4.2 is very difficult for me to follow.  Since the tree is determined given {d} and {h}, it is odd for the authors write of probabilities of constitutents and relative word heights.  If the \beta_{i,k} and p_j^{C(i)} are taken as probabilities, it doesn't look like the distribution over resulting trees will in general be proper.  And I am not even sure how it makes sense given the tree topology. For example, if j is in between i and head(i), then j *must* be in the smallest constituent that contains both w_i and w_head(i).  But head(i) is not even referred to in Equations 4 and 5.  Also note I, thattt can make all the { p_j^{C(i)} } arbitrarily large by setting the bias term to a large positive number, or arbitrarily small by setting the bias term to a large negative number.  I was confused enough after the first paragraph of Section 4.2 that I could not make sense of the second paragraph.  I would recommend a careful and comprehensive rewrite of this section.

* in line 6 of Algorithm 1, it looks like the order in which the {d_i} are evaluated will matter for the resulting tree, and the i should be considered either in the order 1 to len(w) or len(w) to 1.  Clarification needed.

* The only time that the "gap" argument in Algorithm 1 is mentioned in the paper is at the bottom of line 6, where they say they set gap = 0.1.  How was this determined?  Was this value optimized on held-out data (not the test set)?

* is tie-breaking on distances or heights ever needed for Algorithm 1?  If so, how is tie-breaking done -- randomly?

* in section 4.2, are \tau and b learned?

* Section 5.4: "We also observe that StructFormer converges much faster than the standard Transformer model" -- can you report this more precisely?

Errors:

* in Algorithm 1, the "gap" argument is missing in the call to BuildTree in line 8.

* also in Algorithm 1, lines 12 and 13 seem to be reversed.

* equation 9 seems wrong: I believe that the summand should be p(i -> k) p(j -> k).

---

### Official Review · AnonReviewer1 · 2020-10-29
**Interesting adaptation of PRPN for dependency parsing**

**Rating:** 6
**Confidence:** 4

**Review:**

SUMMARY

The paper extends PRPN with the syntactic height in Luo et al. (2019) to do dependency parsing simultaneously with constituency parsing. The main idea is to exploit constituents to infer dependencies. The heights and distances are learned by a masking scheme similar to PRPN, but the model is a tranformer MLM rather than a recurrent memory network. The proposed model achieves nontrivial parsing performance.


STRENGTHS
- The observation that dependencies can be recovered from syntactic heights given constituents is insightful and motivates a natural joint parsing algorithm (Algorithm 1).

- The new masking scheme that explicitly enforces MLM to be sensitive to dependency structure is interesting.

- The experimental results are interesting. The model outperforms ON-LSTM in constituency parsing and is competitive with classical NLP methods that use gold POS tags in dependency parsing. Perplexity, while less helpful to look at in MLM than in LM, also improves substantially especially with small data.


WEAKNESSES

- Some derivations feel too terse. In the central equations (4) and (5), it is difficult to reason what it means for a word to be higher than the all the boundaries in a constituent. Intuitively this corresponds to the word being in the constituent, but some more rigorous explanation will be helpful.

- The paper is a bit sloppy in writing. The issues I had include: how is Definition 3.1 different from "close to the root node" (and why n-1)? In Algorithm 1 I will get an undefined variable error for k (line 13 should be above 12). How are the attention masks defined in equation (10) used? In fact, I still have difficulty understanding how the MLM is trained, as there is no mention of loss or objective (Section 5.1 is not about MLM: it is about data). I guess it is just standard MLM, but this seems to require some acknowledgement.

- The dependency parsing performance is far behind the current SOTA (neural DMVs), even when we account for the fact that it does not use the POS information. It is still not much better than the left-branching baseline (33.3 vs 41). While this is probably due to the inherent difficulty of dependency induction, it is a bit underwhelming.

---

### Official Review · AnonReviewer2 · 2020-11-02

**Rating:** 5
**Confidence:** 4

**Review:**

This paper proposes a neural network optimized by MLM loss that has inductive bias to be useful for unsupervised constituency and dependency parsing. The core component of the neural net is "parser" that predicts a syntactic height and syntactic distance for each token. Following previous work, the distances yield constituency parse while the heights, in conjunction with the constituency parse yield the dependency tree.

The attention heads of the model focus on different dependency based masks for training.

The experimental results show that the method is effective to train a model in an unsupervised manner using just the raw data to predict dependency and constituency parses and performs better than the relevant baselines.

Moreover, the ablation study shows that the different kinds of proposed dependency masks are important for obtaining good performance.

While the paper claims that Figure 4 shows that attention heads learn to focus on different kinds of masks, I do not see a great difference between different attention heads and none of them seem to be biased toward one particular kind of mask. However, attention heads 1,2, and 5 do look interesting. Further analysis of attention heads would make the paper stronger.

One area where the paper suffers the most is readability. The math is broken at several spots and I am not sure if I understand the training objective (section 4.2) correctly.

-- In the algorithm: line 8 -- the buildtree module seems to be taking different arguments than the procedure defined. The algorithm itself could be more clearly written down (the iterator symbols i is inconsistently used and is very confusing)

-- Section 4.2: Maybe it would be helpful to describe the math with an example. But abuse of notation makes the paper incredibly unclear.

-- -- For example what is p_j^{C_i} and p_{ij}^{ct} -- do they refer to similar quantities?

-- -- How can a "token" w_j be a constituent spanning w_i and w_{head_i}?

-- --Equation 9 doesn't look right. Equation 8 should be explained more clearly.

These examples of inaccessible writing (Algorithm and section 4.2) make the core part of the paper difficult to understand and hence I think this paper can be better with another round of thorough revision.

---

### Author Response · Authors · 2020-11-25
**General Response: Paper Revision Updated**

Dear ACs,

Thank you again for the efficient handling of our manuscript. We have made changes in the revision according to the comments from the reviewers. More specifically, we made following changes:

1)To address the clarity issue:
We rewrote model description sections (3 and 4). And they reflect our recent updates to the model too. We also rewrote the experiment sections to provide more detail on the experimental setting and results.

2)To address the comparability issues:
Following the settings proposed in previous unsupervised constituency parsing papers, we retrained our model on the PTB dataset and evaluated it on the WSJ test set. Under this setting, our model achieves unsupervised constituency parsing score 54.0 (UF1) and unsupervised dependency parsing score 46.2 (UAS).

We sincerely hope that reviewers could spend some time to read the new version of our paper.

Thank you for your time and help!
Best, Authors

---

### Decision · Program_Chairs · 2021-01-07
**Final Decision**

**Decision:**

Reject

**Comment:**

This paper presents a novel approach to grammar induction. Like older work by Klein and Manning, the paper finds benefit in jointly inducing both constituency and dependency structure. However, unlike most approaches to grammar induction, the model is not generative -- rather, it is a transformer-based architecture that is trained to optimize a masked language modeling objective. The resulting parses appear to beat non-trivial baselines, but direct comparisons with several relevant state-of-the-art systems are not drawn. Reviewers overall found the approach interesting and novel. However, nearly all reviewers raised serious concerns about experimental comparisons with related work and brought up several missing state-of-the-art baselines that, like the proposed system, do not require gold POS. Reviewers also pointed out issues with clarity in several sections. In rebuttal, authors provide a substantial update to the original draft. So substantial that all reviewers mentioned in discussion that the new draft would effectively require an entirely new review. While I applaud authors for the substantial revisions, and while ICLR guidelines do not explicitly limit the amount change to a draft allowed in rebuttal, in this case the revisions are sufficiently drastic that I agree with reviewers that a new review process is required. Thus, I recommend rejection but strongly encourage authors to resubmit.